# Growth Process and Mortality of *Sasa borealis* Seedlings over Six Years Following Mass Flowering and Factors Influencing Them

**DOI:** 10.3390/biology14050516

**Published:** 2025-05-07

**Authors:** Hanami Suzuki, Hisashi Kajimura

**Affiliations:** Laboratory of Forest Protection, Graduate School of Bioagricultural Sciences, Nagoya University, Furo-cho, Chikusa-ku, Nagoya 464-8601, Japan

**Keywords:** abiotic factors, biotic factors, dwarf bamboo, emergence, forage, mast seeding, morphology, regeneration, *Sasa borealis*, seedling

## Abstract

Bambusioideae, encompassing bamboo and dwarf bamboo, is a subfamily of clonal plants reproducing through rhizome-type nutritional reproduction. The duration of nutritional reproduction varies from three years to more than 120 years, depending on the species. This process is characterized by synchronous sexual reproduction among individuals within a region, followed by mortality after sexual reproduction. From 2016 to 2017, *Sasa borealis* exhibited sexual reproduction (mass flowering and mast seeding) in northeastern Aichi Prefecture, central Japan. This is a dwarf bamboo species that grows on forest floors. We investigated the regeneration process of *S. borealis* from seed dispersal to germination and the growth of seedlings while considering biotic and abiotic factors. This study clarified the timing of seedling emergence and the factors influencing their number. We observed that the growth and morphological characteristics changed because of differences in the forest floor environment and foraging on parts of the seedlings. The findings regarding regeneration following sexual reproduction in this study provide the first step toward quantifying the regeneration process and timeline for environmental recovery after severe disturbances to forest ecosystems.

## 1. Introduction

Bambusioideae is a subfamily of clonal plants that reproduce through rhizome-type nutritional reproduction. In this process, each individual (genet) extends its underground stem and produces multiple aboveground stems (culms, ramets) from the stem [1]. The duration of nutritional reproduction can range from three years to more than 120 years, depending on the species. The process is typically characterized by semelparity, where extended nutritional reproduction is followed by synchronous sexual reproduction, leading to the death of the individual [1,2]. Dwarf bamboo, primarily consisting of smaller species within the Bambusioideae subfamily, initiates a new generation through seed germination resulting from sexual reproduction. However, seed dormancy periods vary in duration, and the timing of seed germination differs among species [3,4,5,6,7,8]. Similar to germination timing, post-emergence seedling growth differs among species in terms of annual growth rate and underground stem elongation duration [3,4,5,7,9,10,11]. These speeds provide important information for clarifying their effects on forest ecosystems. Specifically, they influence the time required for recovery to the original condition prior to sexual reproduction.

In general, recovery following sexual reproduction in plants (i.e., seed germination and seedling regeneration) is influenced by biotic and abiotic factors. During the pre-germination stage, seeds are subject to predation by seed-eating rodents, insects, and other animals, which subsequently influences the number of germinations [12,13]. Seeds of dwarf bamboo and other Bambusioideae are subject to predation by rodents and birds [14,15,16,17,18], indicating that this predation pressure affects later regeneration. Germination can only occur under moist conditions for non-dormant seeds, whereas dormant seeds require abiotic conditions, such as temperature and light [19]. Germination experiments conducted on dwarf bamboo under controlled environmental conditions have shown that temperature and humidity affect germination rates [20].

A biological factor influencing seedling mortality after seed germination is foraging by mammals, with ungulates, rodents, and other animals affecting seedling survival rates [21,22,23]. Dwarf bamboo, in a nutritionally reproductive state, serves as a major food resource for deer, particularly during winter, when food is scarce [24]. It is widely used by mammals, including foraging by field mice and Japanese macaques [25,26,27]. Seedlings may become targets of their foraging and face mortality. Overforaging can sometimes cause morphological changes in dwarf bamboo or result in community degradation [24,28,29,30,31,32,33]. This morphological change has been observed in seedlings [34] and must be carefully monitored. The presence of homologous plants also needs to be considered. Simultaneous production of seedlings at high density may lead to competition for resources, resulting in self-thinning, as has been observed in dwarf bamboo [4]. Conversely, abiotic factors include soil conditions, temperature, precipitation, and solar radiation [35,36]. Experiments on the manipulation of growth conditions have shown that the amount of sunlight affects the growth rate of dwarf bamboo; differences in morphology, such as higher culm height in darker areas, have been observed [11,37]. However, responses to these factors and their growth are expected to differ among species and require verification.

From 2016 to 2017, synchronous sexual reproduction (mass flowering and mast seeding) of *Sasa borealis* occurred in northeastern Aichi Prefecture, central Japan. This is a dwarf bamboo species that grows on the forest floor. In Japan, it is distributed mainly on the Pacific Ocean side from the eastern part of Hokkaido to Honshu, southwest Japan, Shikoku, and the southern part of Kyushu, as well as on the Sea of Japan side of the Korean Peninsula [38]. It is highly resistant to freezing, but in regions with snow cover, it appears on steep slopes and south-facing areas with little snow cover to avoid snow pressure [38]. Although the sexual reproduction of *S. borealis* has been reported in Japan and the Korean Peninsula since the 2010s [39,40,41,42], knowledge of its regeneration is limited [43,44]. The loss of vegetation that previously covered the forest floor constitutes a major disturbance to forest ecosystems. Therefore, the progression of future regeneration and the timeline for recovery should be quantified.

In this study, we examined the *S. borealis* regeneration process from seed dispersal to germination and seedling growth over six years following mass flowering and mast seeding events. Simultaneously, we investigated the biotic and abiotic factors affecting the seedlings to determine their influence on germination, growth, and mortality. The study seeks to provide detailed insights into the timing of seedling emergence, long-term growth patterns, and overall *S. borealis* regeneration dynamics in its natural habitat.

## 2. Materials and Methods

### 2.1. Study Site

This study was conducted in the Takatokke District of the Nagoya University Forest (in the Inabu Field affiliated with the Graduate School of Bioagricultural Sciences, Nagoya University, 35°13′03.0″ N, 137°34′22.0″ E), located in the northeastern part of Aichi Prefecture in central Japan. The elevation was approximately 1075 m asl. In 2022, the annual precipitation was 2460 mm, and the average annual temperature was 9.2 °C. The forest featured a patchy distribution of broadleaved forests containing various tree species, such as *Quercus crispula*, *Castanea crenata*, *Padus grayana*, and *Acer* spp. Mass flowering and mast seeding of *S. borealis* occurred at and around the site in 2017. Partial flowering was observed in some areas in the previous year. Two study sites (SF1 and SF2) were established approximately 300 m apart on a ridge.

### 2.2. Setting Up Study Plots

In 2018, four 15 m × 15 m plots (Plots A, B, C, and D) were established at these sites. Plots A and B were located in SF1, whereas Plots C and D were located in SF2, with all plots adjacent to or encompassing streams or wetlands. The criteria for plot establishment included similar forest type, density of dead *S. borealis* culms, and level terrain. Plots A and C were enclosed by polyethylene netting (height: 1.2 m, mesh size: 13 cm) supported by wooden stakes to keep out medium- and large-sized mammals, such as sika deer (*Cervus nippon*). The net did not affect the movement of small mammals, such as field mice. The combination of Plot A and Plot C was designated as Plot E, while that of Plot B and Plot D formed Plot N. Two types of study quadrats (Quadrats E and N) were placed in Plot E, whereas one type of study quadrat (Quadrat N) was placed in Plot N, with each containing eight quadrats. Each quadrat was 0.7 m × 0.7 m in size. Quadrat E was designated as a field mouse exclusion area. By August 2018, a cage constructed from hexagonal wire netting, measuring 0.7 m in height and featuring a mesh size of 10 mm, was installed above ground, effectively preventing field mice from entering the cage from below. Quadrat N served as a non-exclusion area for field mice, with plastic plates positioned on each of the four corners as markers.

Therefore, three types of operational test sites were established at the study site: Plot E-Quadrat E (EE), which excluded the entry of medium- and large-sized mammals and field mice; Plot E-Quadrat N (EN), which excluded only middle- and large-sized mammals; and Plot N-Quadrat N (NN), which allowed the entry of all mammals.

### 2.3. Survey of Seedlings

#### 2.3.1. Emergence of Seedlings

The number of seedlings that emerged in each quadrat was counted from 2018, and those counted were numbered or marked. Culms were numbered by attaching a twist tie with vinyl tape at the end. Marking was performed by attaching a looped twist tie to the culm. Individuals identified in 2018 were referred to as “2018 seedlings”, and followed by “2019 seedlings” and “2020 seedlings” in subsequent years. Surveys were conducted approximately every two weeks from April to November each year, starting in 2018.

#### 2.3.2. Mortality and Morphology of Seedlings

The survival of individuals numbered or marked in quadrats were checked once every two weeks to monitor the loss of seedlings. Upon confirmation of dead or missing seedlings, the causes and conditions of the seedlings were classified into the following categories: cut (CU), the culm was cut, with the upper part of the cut culm remaining and falling down; forage (FO), the culm was cut, with the upper part of the cut culm being absent, or bite marks or other abnormalities were observed in the twist tie that remained after the individual was not identified; damping off (DO), the entire seedling was wilting; and unknown (UK), not fitting into any of the above categories. Surveys were conducted between April and November each year from 2018 to 2022 and between April and August 2023.

To track the morphological growth of seedlings, the number of culms, leaves, and culm height were recorded for numbered individuals. The number of culms and leaves was recorded every four weeks from April to November each year from 2018 to 2022 and from April to August in 2023. Simultaneously, we recorded the occurrence of foraging. Culm height above ground was measured in November of each year, with measurements taken in September for 2023. For individuals with more than one culm, the maximum culm height was used. In Plots C and D, marked individuals were randomly selected for measurement, resulting in a total of 15 individuals, including the numbered individuals in each quadrat.

### 2.4. Measurements of Biotic Factors

#### 2.4.1. Medium- and Large-Sized Mammals

The effects of medium- and large-sized mammals on the seedlings were investigated using camera trapping. The investigation sites were Plots B and D, which were both unfenced. Two automatic sensor cameras (HykeCam SP2, Hyke Inc., Asahikawa, Japan) were installed within each plot at a height of approximately 120 cm above the ground and operated continuously throughout the year from 2018 to 2023. The sensor camera was set to a low sensor sensitivity to capture three images. The species, dates, times, and confirmed behaviors of the captured mammals were recorded. The obtained data were compared with seedling data from quadrats during the same period, and the foraging activity of medium- and large-sized mammals on seedlings was recorded. When foraging behavior outside the quadrat was observed, the corresponding area was checked and noted in the field.

#### 2.4.2. Field Mice

The effects of field mice on seedlings were investigated using camera trapping. The investigation was conducted at two quadrat sites where field mouse access holes were present. An automatic sensor camera (SG560P-8M, Boly Media Communications Co., Santa Clara, CA, USA) was set up to monitor seedlings in quadrats over the entire year from 2019 to 2023. The sensor camera was set to a low sensor sensitivity to capture video for 10 s following sensor activation. The date and time of field mouse invasions were recorded from videos, and the species were identified as closely as possible. The obtained data were compared with seedling data from quadrats during the same period, and the presence of foraging by mice on seedlings was recorded.

### 2.5. Measurements of Abiotic Factors

#### 2.5.1. Temperature and Humidity near the Ground Surface

The temperature and humidity 20 cm above the ground were recorded using a data recorder (Thermo Leaf, Taisei Fine Chemical Co., Tokyo, Japan). Two locations were established in each plot, and the average data were used to represent the temperature and humidity of each plot. Data were recorded hourly in this setting. The measurement period extended from early April to the end of November of each year, excluding the snow season.

#### 2.5.2. Solar Radiation Intensity

Solar radiation was measured 20 cm above the ground using a simplified solar radiation measurement film (Opto Leaf (Y-1W), Taisei Fine Chemical Co., Tokyo, Japan). Simultaneously, a pyranometer (PYR Solar Radiation, METER Group, Pullman, WA, USA) was set up in the forest to create a calibration curve for the film fading rate. The calibration curve was used to calculate the solar radiation at the location where the film was placed. The pyranometer recorded average solar radiation intensity (W·m^−2^) every hour, and the recorded values were subsequently converted into integrated solar radiation per hour (MJ·m^−2^). The measurement period spanned from June to September each year. The locations were the two opposite corners of the half quadrats, and the average value was used as the integrated solar radiation for that quadrat.

#### 2.5.3. Soil Moisture and Nutrient Content

Soil moisture content and electrical conductivity (EC), which reflects the concentration of water-soluble salts in the soil, were measured using a soil sensor (Delta-T Devices, WET150 Meter, Cambridgeshire, UK). Measurements were conducted at two locations within each quadrat over two days at the same location between October and November 2022. The average value of the two days was used as the measurement value for each location.

#### 2.5.4. Snow Level

The snow coverage was recorded for each year using the time-lapse function of an automatic camera (HykeCam SP2, Hyke Inc., Asahikawa, Japan). Cameras were set up from the beginning of December to the end of April of the following year, and pictures were taken daily at 9:00, 12:00, and 15:00. The recording location was adjacent to each plot in Plots A and C. The date of snow cover was recorded from captured images, and the depth of snow cover was measured using a red-and-white pole positioned within the area of image capture as a ruler.

### 2.6. Data Analysis

#### 2.6.1. Number of Emerged Seedlings

Significant differences in quadrats regarding the presence or absence of field mice were assessed using the Mann–Whitney U-test or Student’s *t*-test to evaluate the number of germinated seedlings each year. We conducted a model analysis using a generalized linear model (GLM) to identify the factors influencing the number of seedling occurrences within each quadrat, with the number of seedlings germinated in the quadrats during 2019 as the objective variable. The model employed a Poisson distribution with a log-link function. The total solar radiation per day, the number of dead culms as an indicator of seed supply, and the presence of field mice in quadrats were used as explanatory variables. Factors were selected based on the best model selection using Akaike’s Information Criterion (AIC). The presence or absence of field mice was used as a dummy variable (exclusion, 0; non-exclusion, 1).

#### 2.6.2. Mortality and Morphology of Seedlings

Mortality rates of seedlings were compared among the different germination years (2018, 2019, and 2020). The rates were calculated annually after germination based on the percentage of the number of deaths to the number of germinations. Total rates during the three years after germination were also determined for each year of germination.

To examine the occurrence of self-thinning as a potential cause of DO, the Pearson product-moment correlation coefficient was used to test the correlation between the number of seedlings that died owing to DO or the rate of mortality owing to DO and the density of seedlings germinated in each quadrat. A positive correlation observed between seedling density and DO rate suggests the occurrence of self-thinning. To determine the relationship between the timing of increased DO and weather conditions, we analyzed the correlation between the number of DO recorded on the survey day and the amount of rainfall over the month preceding the previous survey day using the Pearson product-moment correlation coefficient. The annual precipitation was derived from data collected in the Inabu field. These analyses were performed on the 2019 seedlings.

Comparative analysis of the morphological growth of seedlings was conducted using the Steel–Dwass test or the Mann–Whitney U-test to assess differences in culm height across emergence and growth years. To investigate the factors affecting the growth of culms, a model analysis was performed using a generalized linear mixed model (GLMM) with the average height above ground per quadrat and average growth of culms per year as objective variables. The lme4 package [45] of the R software (version 4.1.2 [46]) was used for the analysis, and the model used a Gaussian distribution with an inverse link function. In the analysis where mean culm height served as the objective variable, the explanatory variables, including total daily solar radiation, near-surface temperature (average, maximum, and minimum from April to October), mean near-surface humidity (from April to October), number of seedlings (November of previous year and November of current year), soil moisture content, soil EC, plot type (E or N), quadrat type (E or N), snow cover (days and maximum depth), annual precipitation, study site (SF1 or SF2), and years since seedlings emerged, were included in random effects. If the average growth for one year was used as the objective variable, years since emergence were added to the above explanatory variables, and the study site (SF1 or SF2) was included in the random effects in the analysis. Plot and quadrat types were set as dummy variables (Plot E, 0; Plot N, 1; Quadrat E, 0; Quadrat N, 1). These model analyses were performed on the 2019 seedlings. For scheduling reasons, a one-year period refers to the period from November of the previous year to November of the following year.

#### 2.6.3. Mortality and Morphology of Foraged Seedlings

The following analyses were conducted using morphological, mortality, and foraging damage data from the same seedlings that were monitored. Foraging damage was categorized into “culm foraging damage” caused by mammals, which foraged from the tops of culms with leaves, and “leaf foraging damage” caused by arthropods, which foraged partially on leaves.

First, to assess the extent of foraging damage that resulted in seedling mortality, the number of surviving and deceased seedlings affected by culm foraging was recorded. The difference in these numbers over one year was tested using a *χ*^2^ test. Residual analysis was used to analyze significant differences between the number of survivors and mortalities. Furthermore, the mortality rate of seedlings affected by culm foraging damage was calculated. Data were analyzed for each year from November of the previous year to November of the current year.

Next, to evaluate the effect of excluding mammals that cause foraging damage, we compared culm heights among EE, EN, and NN for the three patterns of mammal exclusion at different levels. The 2019 seedlings with the largest number of individuals were used for this validation and calculated based on years of growth. Differences between levels were tested for significance using the Steel–Dwass test.

Moreover, to clarify the effects of culm and leaf foraging on seedling morphology, annual growth metrics, such as height above ground, number of culms, and number of leaves, were analyzed based on the presence or absence of foraging damage as well as the timing of the damage—whether it occurred in the current year or in previous years. The annual growth from November of the previous year to November of the current year (September in 2023) was used for height above ground, while the annual change in the number of culms and leaves from August of the previous year to August of the current year was used for the number of culms and leaves. A GLM was used in the analysis, and the model used a Gaussian distribution with an identity link function. The objective variables included the growth of height above ground, change in the number of culms, and change in the number of leaves per year. The explanatory variables were the presence or absence of culm foraging damage in the current year, presence or absence of culm foraging damage in the previous year, presence or absence of leaf foraging damage in the current year, presence or absence of leaf foraging damage in the previous year, interaction between culm and leaf foraging damage, and age of seedlings since emergence. The presence or absence of foraging damage was used as a dummy variable (absence, 0; presence, 1).

All statistical analyses were conducted using R version 4.1.2 [46].

## 3. Results

### 3.1. Emergence of Seedlings

The total number of seedlings that emerged within the surveyed quadrats (n = 24) was 77 in 2018. In 2019, there was a great increase to 1248, followed by a sharp decrease to 33 in 2020, with no seedlings observed in subsequent years. Therefore, the total number of seedlings over the three years was 1358, resulting in an average density of 53.2 seedlings·m^−2^. The first seedlings recorded each year occurred on 17 August 2018, 23 July 2019, and 11 August 2020. The number of newly emerged seedlings peaked at 19 on 26 September 2018, 775 on 12 and 24 September 2019, and 10 on 9 September 2020 (Figure 1). The final newly emerged seedlings in 2018 and 2020 were observed on 5 December and 20 October, respectively, whereas in 2019, emergence continued after 26 November, the last survey day.

In a comparison of the mean number of seedlings (±SD) between Quadrat E and Quadrat N within Plot E across years, Quadrat E exhibited a significantly higher value of 30.3 ± 12.3 (n = 16) than Quadrat N at 20.4 ± 8.5 (n = 16) in 2019 (Student’s *t*-test, 2019; *p* < 0.05, Figure 2). No significant differences were observed in the other years (Mann–Whitney U-test, 2018; *p* = 0.151, Student’s *t*-test, 2020; *p* = 0.473, Figure 2). Appendix A presents the model selection results regarding the number of emerged seedlings in 2019 within quadrats where solar radiation was measured as an objective variable and total solar radiation per day, number of dead culms, and presence of field mice as explanatory variables. The variables included in the most effective model with the lowest AIC values were the total solar radiation and the presence or absence of field mice (Appendix A). The number of emergences decreased with increased total solar radiation in the quadrats, and it was demonstrated that the number of seedlings was higher in the absence of field mice, specifically in Quadrat E.

Emerging culms originated from the underground stems of individuals that underwent sexual reproduction rather than from seedlings in 2018 and 2019. In total, 22 emerged in 2018, only one was alive in 2019, and all died by 2020.

### 3.2. Mortality and Morphology of Seedlings

Figure 3 presents the total number of emerged seedlings, the number of remaining individuals, and the number of dead individuals categorized by cause and condition. The mortality rate on 23 August 2023 was 41.7%. The total mortality rates during the three years after germination were 80.5%, 36.8%, and 51.5% for the 2018, 2019, and 2020 seedlings, respectively. The 2019 seedlings had a much lower mortality rate than the 2018 and 2020 seedlings in the current year of germination (Appendix A).

Regarding seedling mortality, DO due to the wilting of leaves and culms was the most common cause, with a total of 373 individuals over five years, accounting for 65.9% of the total mortalities. Their concentration increased after August, with levels notably elevated in 2022 compared to the other years. Figure 4 shows the correlation between the number of DO, the rate of mortality due to DO, and the density of seedlings that emerged in each quadrat. There was a positive correlation between the density of seedlings in each quadrat and the number of DO (Pearson product-moment correlation coefficient: cor = 0.494, *p* < 0.005), but no correlation with the DO rate (Pearson product-moment correlation coefficient: cor = −0.024, *p* = 0.870). The relationship between the number of DO and total rainfall in the month prior to the check was not significant (Pearson product-moment correlation coefficient: cor = 0.363, *p* = 0.068) (Appendix A).

The second leading cause of mortality was FO, resulting from damage to seedling tops caused by foraging. This damage predominantly occurred during the winter period from November to April each year, accounting for 144 seedlings or 25.4% of the total mortality over the five years.

The following data regarding the seedling morphology were obtained. In November 2018, the mean culm height was recorded at 4.4 ± 1.5 cm (n = 57, mean ± SD) (Figure 5). In November 2019, the mean height of the 2018 seedlings was 6.3 ± 1.4 cm (n = 32), whereas that of the 2019 seedlings was 4.2 ± 1.3 cm (n = 363). In November 2020, the mean height of the 2018 seedlings was 8.3 ± 2.3 cm (n = 20), that of the 2019 seedlings was 7.2 ± 2.2 cm (n = 785), and that of the 2020 seedlings was 4.0 ± 0.9 cm (n = 29). Similarly, in November 2021, the mean height of the 2018 seedlings was 8.2 ± 2.1 cm (n = 13), that of the 2019 seedlings was 8.3 ± 3.2 cm (n = 690), and that of the 2020 seedlings was 5.9 ± 2.5 cm (n = 20). In November 2022, the corresponding measurements were 9.7 ± 2.9 cm (n = 12), 10.3 ± 3.7 cm (n = 577), and 6.5 ± 2.1 cm (n = 14), respectively, while in September 2023, the corresponding measurements were 9.6 ± 3.0 cm (n = 11), 11.1 ± 4.7 cm (n = 512), and 7.8 ± 1.9 cm (n = 13), respectively. By the year of emergence, significantly higher numbers of first- and third-year 2019 seedlings were observed (Steel–Dwass test, *p* < 0.05). Other years exhibited a higher number of 2019 seedlings, although these differences were not significant. Therefore, except for the current year’s seedlings, the 2019 seedlings tended to be better in the growth tests than the preceding and subsequent years’ seedlings. In terms of the number of years since emergence, there was significant growth from the current-year seedlings across all emergence years (Steel–Dwass test, *p* < 0.05), however, the 2018 seedlings exhibited a slowdown from the second year. The results of the GLMM with objective variables of culm height in each year and the amount of growth in a year are shown in Appendix A. Various factors showing positive and negative effects were significantly related to them.

Figure 6 shows the monthly changes in the number of culms per seedling for each year of emergence. The 2018 seedlings comprised fourth-year individuals with two or more culms, the 2019 seedlings consisted of third-year individuals with almost all displaying multiple culms, and the 2020 seedlings included all second-year individuals with multiple culms. The number of culms varied among seedlings; for example, in the 2019 seedlings, the maximum number of culms was 14 in August 2023, with approximately 10% of seedlings having two culms and almost half having five or more culms. Compared to the year of emergence, in August 2023, the 2018 seedlings tended to have a higher proportion of seedlings with two culms compared to the 2019 seedlings. In addition, the maximum number of culms was 14 in the 2019 seedlings compared to 10 in the 2018 seedlings. In contrast, the 2020 seedlings in 2023 exhibited a higher percentage of seedlings with two or three culms than the 2019 seedlings in 2022.

Figure 7 shows the monthly changes in the number of leaves per seedling for each year of emergence. The trend was generally the same, with the lowest number of leaves in April, increasing through August, and then decreasing throughout the following months. The mean number of leaves (±SD) in August, the month with the highest leaf count during the year, was 4.9 ± 2.2 leaves (n = 36) for the 2018 seedlings in 2019. In 2020, the mean number of leaves of the 2018 seedlings was 5.3 ± 2.7 (n = 20), whereas that of the 2019 seedlings was 5.4 ± 2.3 (n = 478). Similarly, in 2021, the mean number of leaves of the 2018 seedling was 4.7 ± 3.8 (n = 18), that of the 2019 seedlings was 5.9 ± 3.1 (n = 394), and that of the 2020 seedlings was 4.8 ± 3.0 (n = 22). In 2022, the corresponding measurements were 6.1 ± 3.3 (n = 14), 6.8 ± 4.8 (n = 351), and 4.3 ± 2.8 (n = 19), respectively, while in 2023, the corresponding measurements were 5.5 ± 2.2 (n = 11), 6.1 ± 4.9 (n = 280), and 4.3 ± 3.2 (n = 14), respectively. The variability among seedlings was large, and those emerging in any given year exhibited minimal or no increase in the mean number of leaves at peak in the year following emergence.

### 3.3. Seedling Foraging and Damage by Mammals and Arthropods

In 2018, FO to two seedlings occurred only within a deer entry period within the quadrat, the surveyed area of seedlings. During other periods of recorded FO to seedlings, Japanese hares (*Lepus brachyurus*) and deer were recorded using the sensor camera. In all cases, both species were recorded during this period, but it was unclear which of the two species was responsible for foraging. Furthermore, hares foraging on seedlings outside the quadrats were observed several times (Appendix A). Foraging marks were readily identifiable due to the presence of culms with their upper parts cut off, resulting from foraging on the leaf-bearing upper parts of the culms. Foraging activity was recorded once (1 April) in 2022 and twice (1 September and 29 October) in 2023 in Plot B. Deer foraging on tree leaves, fallen leaves, and tree seedlings was recorded several times (Appendix A), and their foraging marks were observed. However, *S. borealis* seedlings in the surrounding areas were not foraged. It was observed that they detected the scent of *S. borealis* seedlings but did not engage in foraging activities. Ground digging by wild boars (*Sus scrofa*) occurred in parts of Plot B in 2018 and throughout Plot B in 2022. Although the seedlings fell down after digging, no direct mortality was observed. Moreover, although the cameras at the study sites did not capture any foraging activity, undigested seedlings of *S. borealis* were identified in raccoon dog (*Nyctereutes procyonoides*) feces on three occasions between 2022 and 2023 outside the study sites (Appendix A).

In the quadrats where the sensor camera was set up for *S. borealis* seedlings, field mice foraging on the seedlings were recorded on 29 March and 2 April 2021 (Appendix A). However, the field mouse species could not be identified. Foraging marks were similar to those of hares, and cut-off points for field mice were closer to the ground level than those for hares.

*S. borealis* seedlings were observed with parts of the leaves chipped off. The occurrence period spanned from spring to early summer during the leaf development stage and autumn (Figure 8). Foraging signs were observed in nine seedlings in 2018, 25 seedlings in 2019, 13 seedlings in 2020, 50 seedlings in 2021, 29 seedlings in 2022, and 12 seedlings in 2023, representing small percentages of the total number of seedlings. No arthropods were observed on the leaf surfaces or foraged on the leaves.

### 3.4. Mortality and Morphology of Foraged Seedlings

Figure 9 shows the number of *S. borealis* seedlings that either died or survived after culm foraging damage by mammals and the mortality rate due to culm foraging damage. All culm foraging damage was treated as total damage and not categorized by the foraging mammal species. The total number of seedlings exhibiting culm foraging damage showed no discernible trend; however, the number of seedlings that died as a result of foraging damage decreased significantly from year to year. Mortality rates showed a significant downward trend: 70.8% in 2019–2020, 38.4% in 2020–2021, 11.6% in 2021–2022, and 7.7% in 2022–2023 (*χ*^2^ test and residual analysis, *p* < 0.001).

The average culm height of *S. borealis* seedlings in Plot-Quadrat, with and without mammal exclusion, is presented in Figure 10. Prior to 2021, average culm height was significantly higher in the NN with potential entry of all mammals and lower in the EE with strong exclusion of mammals (Steel–Dwass test, *p* < 0.05). In contrast, the situation was reversed thereafter, and the culm height was significantly higher in the EE group (Steel–Dwass test, *p* < 0.05).

The results of the GLM examining the presence or absence of culm foraging damage and the presence or absence of leaf foraging damage in each *S. borealis* seedling, categorized by the timing of damage occurrence, are presented in Appendix A. The explanatory variables include the aforementioned damage types, whereas the objective variables consist of culm height growth, changes in the number of culms, and changes in the number of leaves over one year. The culm foraging damage observed in the current year had a negative effect on culm height growth. The culm foraging damage of previous years and the culm foraging damage of the current year negatively impacted the culm increase, whereas the leaf foraging damage of previous years and the leaf foraging damage of the current year positively affected the culm increase. For leaf increase, the culm foraging damage of the current year had a negative effect, similar to the number of years since emergence.

## 4. Discussion

### 4.1. Timing of Emergence

The total number of emergent seedlings of *S. borealis* across all surveyed quadrats was highest in 2019, with a small number of seedlings recorded before and after that year (Figure 1). The earliest observed emergence in each year occurred in late July 2019, with peaks consistently observed in September across all years, followed by ongoing emergence until approximately December, except for 2020. No seedlings emerged after 2021, indicating that the peak of the outbreak occurred two years after seeding (in 2019), with a limited number of seedlings emerging in the years preceding and following that period. The timing of emergence predominantly occurred in the late summer. The peak emergence two years after seeding was the same as that of *S. borealis* in other regions that experienced mass flowering in the same year [44]. This indicates that the dormancy period of *S. borealis* is mainly two years, with rarely one or three years for some individuals. There are a few patches of *S. borealis* that did not flower in 2017 within the same region (Suzuki, unpublished data), which may be related to such a gap in germination time.

Different periods have been reported for dwarf bamboo species in Japan, specifically regarding seedling emergence after mast seeding, as documented in previous studies. *Pleioblastus argenteostriatus* is a non-dormant species that emerges in June or July of the same year following seed dispersal in early to late June [3]. *Sasa jotanii*, a non-dormant species, emerges simultaneously in early July of the same year following its flowering period from March to April [38]. Conversely, *Sasa senanensis* emerges in May or June of the year following seeding, with its dormancy and low-temperature requirements confirmed [9]. *Sasa tyuhgokensis* emerges from June to October of the year following seeding [20]. Moreover, in *Sasa spiculosa*, emergence occurs after two years of seeding, and germination experiments have verified the long dormancy of the same species [8]. As for *Sasa kurilensis*, there are different reports of germination the following year [4] and two years later [10]. Dormancy occurs not only in dwarf bamboo but also in a group of bamboo species that grow in temperate zones [47,48]. As described above, the timing of the emergence of species belonging to the subfamily Bambusioideae and species of the same genus as *S. borealis* varies. *S. borealis* merged two years after seeding; however, its timing, peaking in September, was later than that of the other species. In deciduous forests, the photosynthetic efficiency of young leaves is highest prior to canopy closure, making early emergence beneficial for subsequent survival and growth [49]. Conversely, plant seedling mortality is generally attributed to early spring frost damage, summer drought, and reduced soil moisture [35,50,51]. *Quercus serrata* seedlings are believed to have a phenological escape strategy of delaying emergence from the leaf-expanding stage of adult trees to escape foraging damage by young leaf-eating insects descending from the canopy [52]. Therefore, both biotic and abiotic factors are involved in plant emergence. At this study site, forest floor environmental data indicated days when minimum temperatures were below 0 °C until approximately early May (Appendix A). Humidity near the ground surface fluctuated substantially before July, with a daily minimum humidity below 50%; however, when emergence began, humidity changed and remained stable and high throughout the day (Appendix A). The leaf-expanding period of *S. borealis* is from April to May [53]. These factors indicate that *S. borealis* has adapted and evolved to emerge in late July or later in terms of drought and phenology rather than photosynthetic efficiency.

### 4.2. Factors Limiting the Number of Emerged Seedlings

Model selection using GLM indicated that total daily solar radiation and the presence of field mice were factors affecting the number of emerged seedlings. Specifically, the number of seedlings was higher in quadrats with less solar radiation and in quadrats with no field mice (Appendix A). In germination tests with *Arundinaria gigantea*, a species of dwarf bamboo, seed survivability and germination rates decreased under high light, whereas germination occurred without issue in shaded environments [54]. Storage and germination tests with *S. veitchii* seeds indicated that germination rates drop significantly after 25 days or more of dry storage [20]. Evapotranspiration from soil is highly correlated with the amount of solar radiation [55]. Assuming a relationship between soil drought and the amount of solar radiation, the lower number of seedlings that emerged in quadrats with higher solar radiation in this study may indicate that the germination rate was adversely affected by drought.

Seed predation by rodents and seed-eating birds has been observed in field Bambusioideae seeding experiments [20,54]. In addition, seed-feeding tests on *S. borealis* at the same study site have revealed that large Japanese field mice (*Apodemus speciosus*) and small Japanese field mice (*Apodemus argenteus*) exhibit various foraging behaviors, such as predating, removing, and caching [18]. The number of emerged seedlings in 2019 was higher in Quadrat E, which was devoid of field mice (Figure 2). Therefore, it is assumed that the exclusion of field mice in 2018, one year after seeding, allowed the seeds in the quadrats to survive. However, the seeds in open spaces were predated, resulting in differences in the number of emergence events.

### 4.3. Mortality of Seedlings

The mortality rate of seedlings during the five years from the beginning of seedling emergence was 42%. Under natural conditions, as in this study, the survival rates of *S. kurilensis* and other *S. borealis* seedlings five and four years after their emergence were reported as 65% [10] and 47% [44], respectively. In addition, the mortality rate of the 2019 seedlings was lower than that of the 2018 and 2020 seedlings. This may indicate that the environmental factors at germination after two years, e.g., a decrease in standing dead culms and its resulting improvement in light condition [4,56], work favorably for seedling survival.

The most common cause of mortality at the study site was DO. A positive correlation was found between the density of seedlings that emerged in each quadrat and the amount of DO, whereas no correlation was found with the DO rate (Figure 4). Therefore, no tendency currently exists for self-thinning of seedlings (e.g., DO rate increasing as density increases). Next, no correlation was found between the number of DO and total rainfall in the month preceding its assessment. However, in 2022, when heavy rainfall was mainly concentrated in September, the number of DO was particularly high. In contrast, the number of DO after June 2019, the second heaviest rainfall month, and June 2023, the next heaviest rainfall month, was small. This means that periods of intense, heavy rainfall during specific seasons may have increased the number of DO. The effects of heavy rainfall on seedlings after summer may be attributed to root exposure caused by raindrops [57] or the establishment of a moist environment conducive to the growth of soil plant pathogens [58].

The second most common cause of death, FO, was mainly observed in the winter and early spring. Although foraging damage occurs in seasons other than winter, the resistance of seedlings may decrease due to environmental factors specific to winter, such as cold weather, which increases the mortality rate. In terms of mortality rate, the proportion of seedlings with culm foraging damage that later died decreased each year (Figure 9). This suggests that resistance to foraging damage increases as seedlings grow, allowing them to produce new culms despite damage and enhancing their survival.

### 4.4. Morphology of Seedlings

Culm height growth was moderate, and even after five years of emergence, the seedlings were only approximately 10 cm in height (Figure 5). When compared to other *Sasa* species, *S. veitchii* reaches a height of 41.3 cm [11], and *S. kurilensis* attains 25.3 cm [10] in third-year seedlings. In addition, *Sasa tsuboiana* measures approximately 20 cm in second-year seedlings [5]. Notably, size varies from site to site. These measurements exceed those of *S. borealis*. The heights of mature *S. veitchii*, *S. tsuboiana*, and *S. kurilensis* are 2 m [11], 2.5 m [5], and 3 m [10], respectively. These measurements are approximately the same or higher than *S. borealis*, which is estimated to be 1–2 m [38]. However, despite this comparison, the growth rate of *S. borealis* appears to be slower. The mean culm height of *S. borealis* seedlings was 5.7 cm in the first year of development and 7.3 cm four years later [44]. In addition, the mean culm height after 5 years of seeding was approximately 10 cm [43]. Therefore, the slow growth of *S. borealis* is a defining characteristic of this species. In addition, growth may be affected by the timing of germination as well as the mortality rate. A comparison of the growth among seedlings from different germination years over the same number of growing years showed a slight trend toward better growth for seedlings germinated two years after seeding (Figure 5). Different germination years have different surrounding environments. For example, the amount of standing dead culms decreases with passing years [4], possibly resulting in an advantage for different germination years. This may be one of the reasons why *S. borealis* germinates two years after seeding.

To clarify the factors affecting the culm growth, a GLMM was employed with culm height and annual growth as objective variables. The analysis showed that the culm grew better at higher temperatures, and soil moisture content adversely affected its growth (Appendix A). The negative effect of seedling density at the end of the previous year and the positive effect of seedling density at the end of the current year may indicate that the density effect suppressed the amount of growth. In addition, the reduced density resulting from seedling mortality due to various factors during the year may have created conditions suitable for growth. Plot N or Quadrat N negatively affected the growth level, suggesting that mammals could enter these open sites. Differences in the culm height of seedlings among operated sites, with and without mammal exclusion, showed a strong tendency: heights were greater at sites where all mammals were excluded and smaller at sites where mammals were present over time (Figure 10). This may be due to mortality from foraging damage. As described above, early seedlings exhibited a high mortality rate following culm foraging damage, resulting in an inability to accurately reflect the effects of foraging damage. However, as the number of seedlings that survived after foraging damage increased, data from these damaged seedlings were incorporated, allowing the effects of foraging damage to become more apparent. Foraging by herbivores generally causes a reduction in the size of dwarf bamboo, a trend also observed in seedlings [34,59,60]. This study indicates that *S. borealis* seedlings are dwarfed by foraging, with this process commencing approximately three years after seedling emergence.

With respect to the number of culms, new culms of seedlings began to appear around May, coinciding with the period when trophically reproductive *S. borealis* begins [53]. The maximum number of culms per individual was 14, and nearly half of the seedlings had more than five culms (Figure 6). The number of culms per individual of other *Sasa* species varies greatly among species. *S. kurilensis* averages 1.59 culms per third-year seedling [4], *S. veitchii* averages 79.7 [11], and *S. senanensis* averages approximately 30 [9]. The observed differences may be attributed to distinct types of underground stems [61,62], suggesting that leptomorph rhizome systems, such as those of *S. tyuhgokensis* and *S. senanensis*, promote a more rapid increase in the number of culms compared to the combined rhizome systems of *S. borealis* and *S. kurilensis*. However, the increase in the number of leaves was modest. This may be associated with the tendency of dwarf bamboo to shed leaves other than those from the current year [63], whereas the annual increase in the number of culms remains minimal.

The effects of non-mortal foraging damage on seedling morphology indicate that culm foraging damage negatively impacts culm growth, and leaf foraging damage had no negative or positive effect (Appendix A). In addition, culm foraging damage negatively affects the increase in culms and leaves, whereas leaf foraging damage positively affects the increase in culms. This suggests that compensatory growth, which is an increase in the number of culms, occurs only when leaf foraging is damaged. Previous studies on other *Sasa* species have confirmed that *S. veitchii* seedlings can reduce culm size and produce culms and shoots when subjected to deer foraging [60]. *Sasa nipponica*, when not in the seedling stage, diminishes in size due to deer foraging, while simultaneously enhancing its tolerance by increasing the number of leaves and culms [28]. In this study, the response of *S. borealis* was found to be consistent only with dwarfing resulting from culm foraging damage and not with an increase in culms or leaves. Moreover, past culm foraging damage had a negative effect on the increase in culms, and past leaf foraging damage had a positive effect on the increase in culms. This suggests that the effect of feeding damage on morphology is long-lasting.

### 4.5. Foragers of Seedlings

A camera survey conducted on medium- and large-sized mammals identified hares as foragers of *S. borealis* seedlings. The leaves of both ordinary and dwarf bamboo are usually located high above the ground, making them less accessible as food sources for hares. However, hares forage on low-regenerating bamboo that appear after the flowering of *Phyllostachys nigra* [64]. Therefore, hares utilize Bambusioideae within their foraging range. *S. borealis* seedlings were located within their foraging range, enabling foraging activity.

*S. borealis* seedlings were not targets of foraging by deer at present because no direct foraging behavior of seedlings was observed, and deer were noted to recognize the seedlings without engaging in foraging activities. *S. borealis*, during nutritional reproduction, is subjected to strong foraging pressure from deer, which in certain regions causes morphological changes or even a decline in colony size [24,29,30,31,32,33]. Therefore, certain differences between *S. borealis* seedlings and *S. borealis* adults determine deer foraging behavior. Regarding plant size, deer had minimal impact on tree seedlings smaller than 20 cm in size [65]. At present, the mean culm height of *S. borealis* seedlings is approximately 10 cm, and the reason for not foraging is that plants near ground level are not foraging targets. However, in this study, it was observed that plants recognized and smelled seedlings but did not engage in foraging, indicating that differences in chemical composition may be factors distinct from seedling morphology. Plants produce various secondary metabolites, such as terpenes, phenolic compounds, and nitrogen-containing compounds, to avoid predation by herbivores, which has proven effective [66,67,68]. The presence of repellent substances in Bambusioideae remains uncertain. However, limonene, a terpene found in conifers, is known to repel deer and is also present in plants belonging to the same Poaceae family as Bambusioideae [69,70]. It has also been detected in moso bamboo (*Phyllostachys edulis*) [71]. If differences in chemical composition are factors in non-foraging behavior, future studies should investigate the presence of other secondary metabolites that are effective in deterring deer.

In the camera survey targeting field mice, foraging on seedlings by field mice was observed only on two occasions. Although the field mice continued to enter the target quadrat at other times, they did not forage on the seedlings. Therefore, the foraging behavior of *S. borealis* seedlings was limited to a few individuals and exhibited seasonal patterns. Three species of field mice have been observed at the study site: large Japanese field mice, small Japanese field mice, and Smith’s red-backed voles (*Eothenomys smithii*) [72]. Large Japanese field mice were excluded based on the body size of the images taken. Smith’s red-backed vole forages on tree seedlings [73], and DNA metabarcoding analysis indicates that small Japanese field mice feed on dwarf bamboo [26]. Therefore, these two species are proposed as possible species.

Although not captured by the camera, it was verified that raccoon dogs foraged on the seedlings. Previous studies of raccoon dog feeding habits have indicated that they forage for family Poaceae grasses and leaves [74,75]. Bamboo foraging by the family Canidae, to which raccoon dogs belong, has also been recorded [76]. It has been suggested that mammals belonging to the order Carnivora do not forage for plants as a source of nutrients, but that their leaves may help expel intestinal parasites [76]. The foraging activity of the raccoon dog remains ambiguous, whether it is anticipated or accidental. However, it is a mammal that warrants continued observation due to its role as a forager in the regeneration of *S. borealis*.

Leaf foraging by arthropods occurred in only a small percentage of the total seedling population, and no mortalities were directly caused by leaf foraging. Specialist insects that forage on parent trees also target their seedlings and threaten their survival [77]. Therefore, the presence of parent trees negatively affects the next generation. Simultaneous dieback of *S. borealis* may have led to the extinction of its specialist insects and subsequently decreased the foraging rate of seedlings.

## 5. Conclusions

By following up on *S. borealis* after its mass flowering and mast seeding events, we obtained detailed observations on the emergence and subsequent growth of seedlings. The peak timing of seedling emergence occurred two years after seeding, while the number of seedlings was adversely impacted by the amount of solar radiation and the presence of seed-eating field mice. Seedlings grew slowly, and their morphology changed because of differences in the forest floor environment, such as high temperature and low soil moisture, and foraging on parts of the seedlings; a tendency toward dwarfing was evident. It is anticipated that *S. borealis* requires a substantial duration to return to its presexual reproductive stage. Hares and field mice were the mammals that generally foraged on seedlings, whereas deer were rarely seen foraging. Future investigations should focus on understanding why deer did not forage on *S. borealis* seedlings. In addition, it is needed to investigate the reason why *S. borealis* germinates after two years of seeding. In general, the prolonged pre-germination period is disadvantageous, as it exposes seeds to the risk of predation during that period. However, the results of this study indicated the probability of advantages, particularly regarding mortality and subsequent growth. For that, the cause of wilting should be determined, which was the primary cause of mortality among seedlings.

## Figures and Tables

**Figure 1 biology-14-00516-f001:**
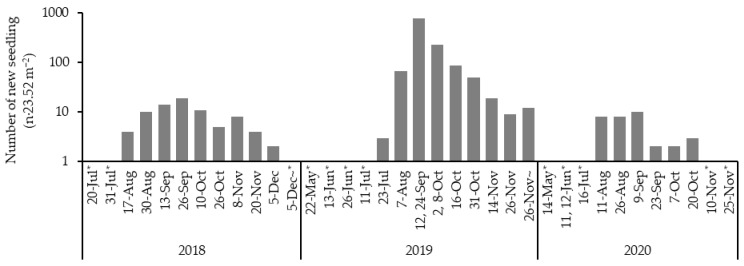
Survey date and the number of newly identified seedlings. Asterisk (*) indicates days when there were no seedlings.

**Figure 2 biology-14-00516-f002:**
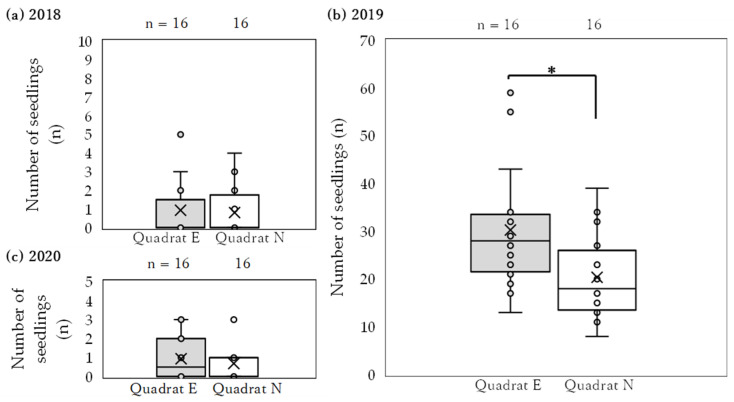
Difference in the number of emerged seedlings of Quadrats E and N within Plot E in each year. Asterisk (*) indicates a significant difference (Student’s *t*-test, *p* < 0.05). Quadrats E and N were designated as field mouse exclusion and non-exclusion areas, respectively.

**Figure 3 biology-14-00516-f003:**
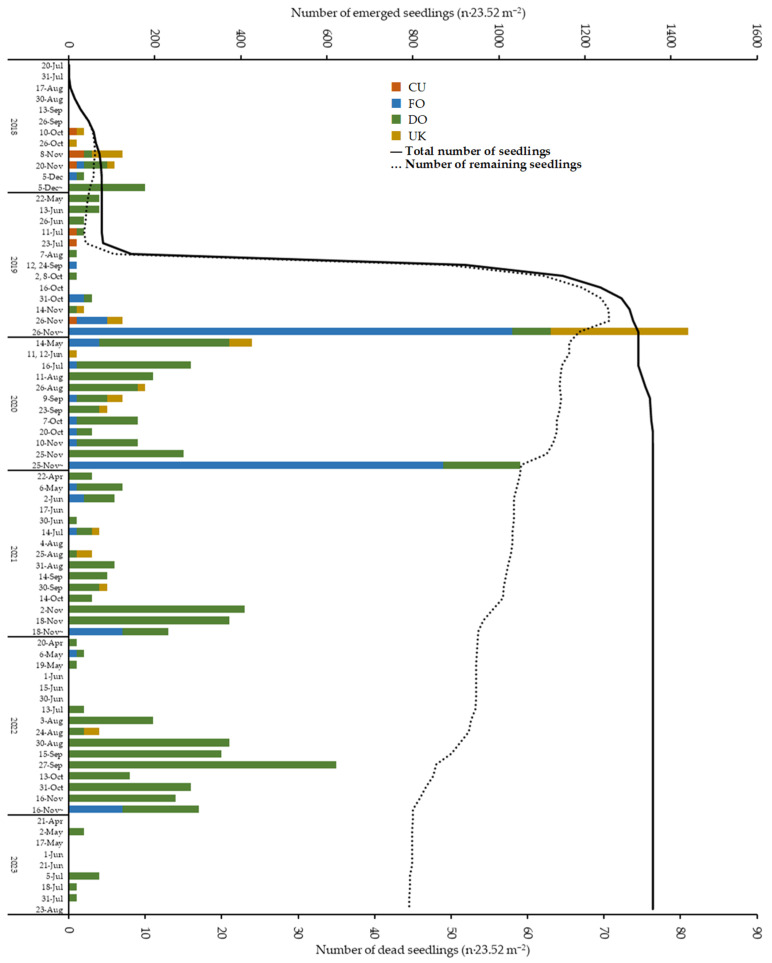
Total numbers of emerged seedlings and numbers of remaining and dead individuals (the latter by cause and condition). CU: the culm was cut, with the upper part of the cut culm remaining and falling down; FO: the culm was cut with the upper part of the cut being absent, or bite marks and other abnormalities were observed in the twist tie that remained after the individual was not identified; DO: the entire seedling was wilting; UK: not fitting into any of above-mentioned categories.

**Figure 4 biology-14-00516-f004:**
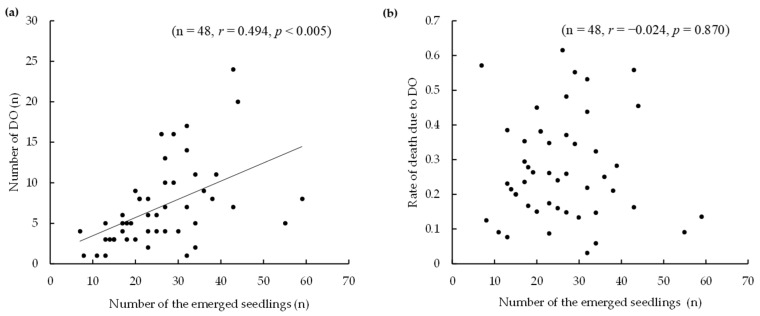
Correlation between (**a**) damping off (DO) numbers or (**b**) DO-related death rate with the number of emerged seedlings in each quadrat.

**Figure 5 biology-14-00516-f005:**
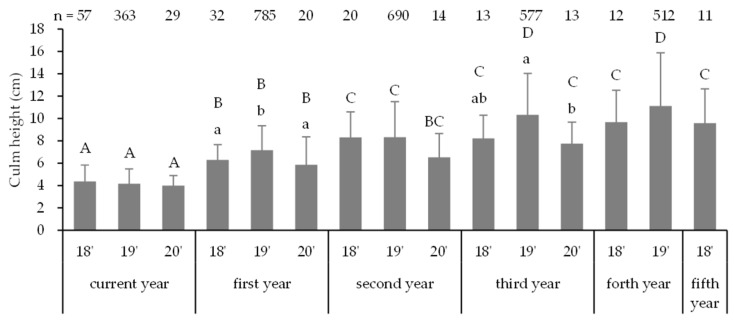
Mean culm heights of the seedlings emerged in 2018, 2019, and 2020. Different capital and lowercase letters indicate significant differences between the number of years since emergence and emergence years, respectively (Steel–Dwass test, *p* < 0.05).

**Figure 6 biology-14-00516-f006:**
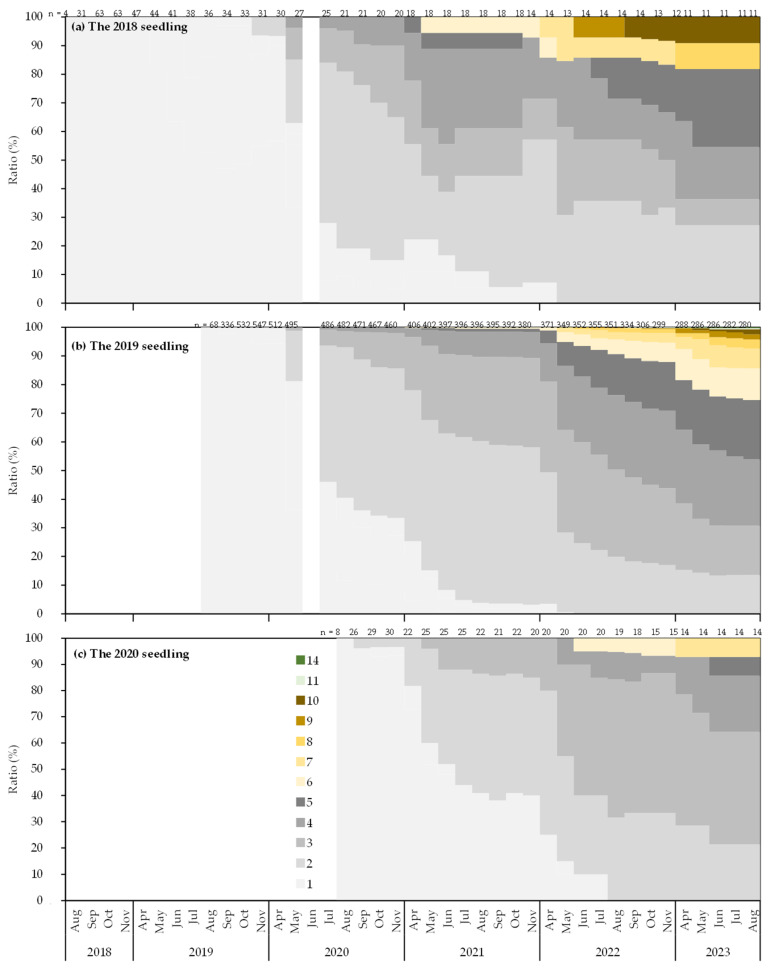
Monthly changes in the ratio of culm numbers per seedling for each year of emergence.

**Figure 7 biology-14-00516-f007:**
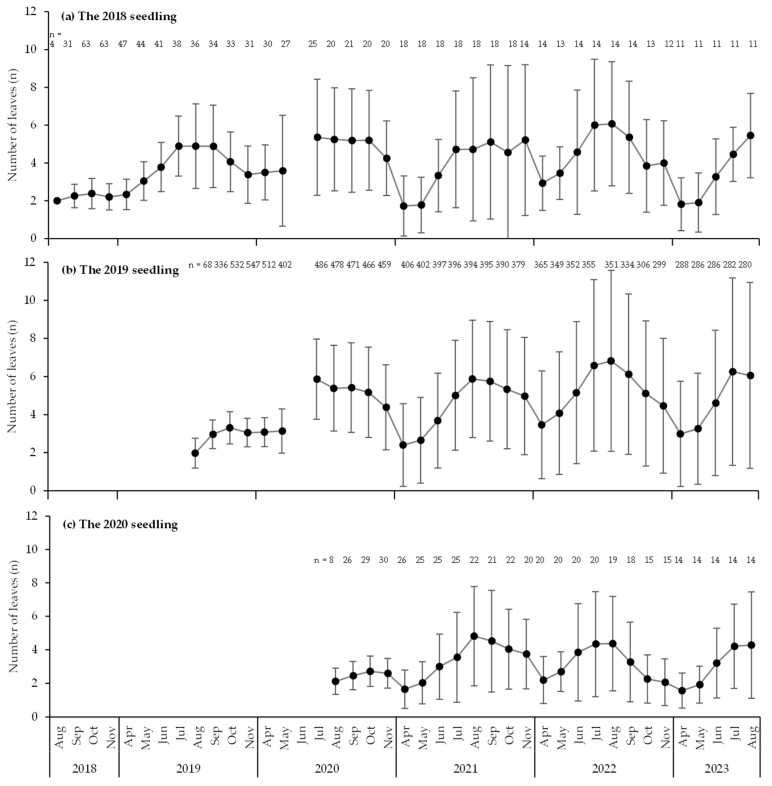
Monthly changes in leaf numbers per seedling for each year of emergence.

**Figure 8 biology-14-00516-f008:**
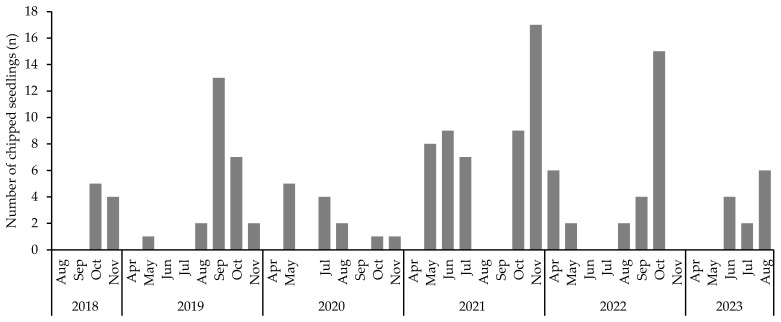
Seedling numbers with parts of the leaves chipped off.

**Figure 9 biology-14-00516-f009:**
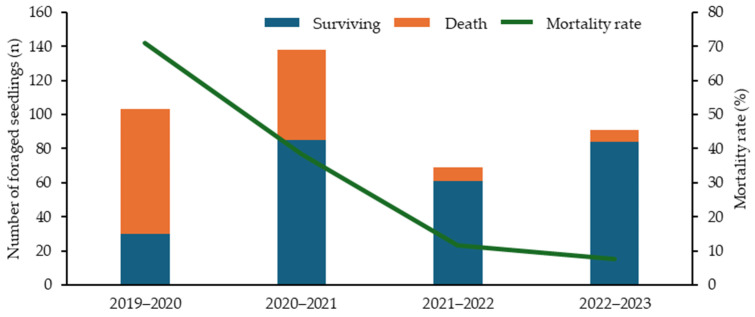
Numbers of seedlings that died or survived after the culm foraging damage by mammals and the corresponding mortality rates.

**Figure 10 biology-14-00516-f010:**
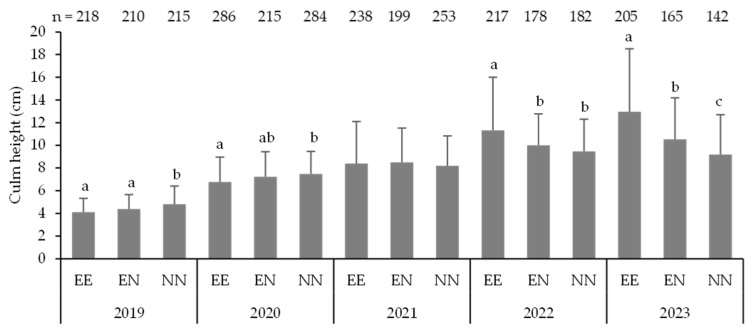
Average culm height of seedlings in Plot-Quadrat with the exclusion and non-exclusion of mammals. Measurements were taken in November each year from 2019 to 2022 and in September 2023. EE: middle- and large-sized mammal and field mouse entry excluded; EN: only middle- and large-sized mammal entry excluded; NN: all mammal entry allowed. Different letters indicate significant differences between the locations (Steel–Dwass test, *p* < 0.05).

## Data Availability

The original data presented in this study are available in FigShare at doi:10.6084/m9.figshare.28104995.

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
