# Peer review of "Growth Process and Mortality of Sasa borealis Seedlings over Six Years Following Mass Flowering and Factors Influencing Them"

_biology, 2025, doi:10.3390/biology14050516_

Round 1
Reviewer 1 Report
Comments and Suggestions for Authors
The flowering behavior of bamboo species is very specific. Because bamboo is an important component of many ecosystems, the research on its flowering behavior is an important ecological topic. However, knowledge of its flowering events of bamboos is unclear because in most cases, the flowering cycle is long, and it is not possible to predict when and where they flower. Thus, this manuscript, which describes detailed records of the seedling development of Sasa borealis after widespread flowering, is worthy of publication as a paper. However, data presentation and discussion is rather insufficient and need further improvement to be completed as a paper.
- It is rare to have such a detailed record of seedling emergence timing as in this study. So, it should make further examination of the advantages and disadvantages of earlier or later emergence timing. Such examination would enhance the value of this paper. As indicated in this manuscript, bamboo seeds are susceptible to feeding damage by rodents in the soil. Nevertheless, why do bamboo seedlings emerge over a very long period? This question should be examined more empirically.
For example, you have verifiable data on the relationship between germination time (germination year and/or those that germinate earlier or later in the same year) and survival rate, the relationship between germination time and growth (size of corresponding seedling in the following year), and the relationship between germination time and the percentage of DO. After such examination, the discussion of “Timing of Emergence” in Disucussion would be valid.
2. A comparison of seedling size among age per year of occurrence (cohort) is shown in Figure 3. However, as already reported (for ex. Nakashizuka 1988 or Makita 1992), the light conditions of the forest floor should change (brighter year by year) because the dead culms of the flowered population are gradually decreased year by year. Thus, the light environment of the 2018 and 2019 cohorts should be different. In this study, the changes in environmental conditions after Sasa death are rarely considered, I feel. It is not appropriate to compare seedling size by seedling age by year of occurrence without fully examining the changes in the environment after the death of the Sasa. The way the data presentation and the content of the discussion should be reconsidered. Consideration on annual changes in the growing environment is indispensable.
For example, on Fig.3, the size of 2yr seedlings of 2018 cohort is similar to 1yr seedlings of 2019 cohort, and 3yr seedlings of 2018 cohort is similar to 2 yr seedlings of 2019 cohort, which indicates the changes of environmental factors after Sasa death are probably important for seedling growth.
3.In Discussion, “Timing of emergence”, it is emphasized that the 2018 emergent seedlings are due to the partial flowering that occurred in 2016. However, I cannot find any description on the flowering in 2016 in this ms. What are the grounds of the argument? Some reports (ex. Mizuki 2014) represented that seedling emergence in the case of partial flowering is rare. I don’t think there is a reason to deny the possibility that this is due to the 2017 fruiting. As you discuss, the emergence timing of bamboos is variable. The argument should be rearranged
4. L 512: Regarding the timing of Sasa germination in the existing literature, S.kurilensis is described as germinating in the second year, but several papers have shown that it occurs in the year following fruiting( Nakashizuka 1988, Makita 1992, Makita et al. 2004). So the statement that Sasa kurilensis is a two-year germinating species is incorrect.
- L574: By comparison with other genus bamboos, Sasa is described to have a low mortality rate. However, compared species, P arenteostriatus have non-dormant seeds, so the initial density of this species may be very high compared to Sasa species. Furthermore, for D. membranaceus, it is a tropical species, and the growing environment is quite different from Sasa. The comparison should be made under the right conditions, and it would be meaningless to compare only numerical values under different conditions.
6: There is a very long discussion. As mentioned above, there are some discussions without sufficient evidence. As for foraging, what is the most important finding in this research? The authors should make re-arrangements for the entire discussion and make a more compact argument put together, based on accurate descriptions and clarification of what new findings should be claimed in this paper.
7.In Conclusion: L724 Although the authors wrote “ emergence after two years was advantageous, particularly regarding mortality and subsequent growth”, I do not think that the results are sufficient to show that mortality and growth are advantageous. The authors would like to see improvements based on the above comments.
- Y axis of Fug 5 is misprinted.
Author Response
Thank you for extending us the opportunity to submit a revised draft of our manuscript titled, “Growth process and mortality of Sasa borealis seedlings over six years following mass flowering and factors influencing them,” for consideration for publication in Biology. We deeply appreciate the time and effort invested by you in providing insightful feedback to enhance the quality of our manuscript. We are sincerely grateful for your valuable suggestions and guidance. It is with great pleasure that we resubmit our article, incorporating the changes that reflect the insightful suggestions. We hope that our edits, along with the responses provided with a pdf file, satisfactorily address all the issues and concerns raised by you.

Reviewer 2 Report
Comments and Suggestions for Authors
The authors have done a lot of work on interesting topics. However, there are a number of questions and comments.
In the Introduction, the authors elaborate on the characteristics of the ecological relationships of the studied S. borealis, but practically do not cover the issues related to the geography of the distribution of this species. Could the authors tell us a little about the distribution range of representatives of this species and the climate features of the regions where it occurs (intensity of insolation, temperature, precipitation, etc.)? Also, could the authors clarify whether there are any bacterial or other infections that reduce the number of these plants?
It is not entirely clear from Figure 1 whether measurements were not carried out in principle on a number of these dates (then perhaps they should be excluded) or whether the data obtained is so much lower than the maximum values that they simply cannot be seen (in this case, the table would probably be more informative).
In Figure 2, all the inscriptions should be duplicated in English.
In Figure 3, the labels at the bottom are almost unreadable.
Section 3.3. Search for food for seedlings and their damage by mammals and arthropods. Have there been any deaths from bacterial infections or shellfish at the sites studied?
I recommend making the conclusions section a little more specific. For example, you should not just write that abiotic and biotic factors are limiting the growth and appearance of seedlings, but specifically list which factors from each group have an impact on this.
Author Response

(The authors gave the same response as above.)

Round 2
Reviewer 2 Report
Comments and Suggestions for Authors
The authors have made the necessary changes